


# Invited Perspective: Building sustainable and resilient communities - Recommended actions for natural hazard scientists

Joel C. Gill[1], Faith E. Taylor[2], Melanie J. Duncan[3], Solmaz Mohadjer[4], Mirianna Budimir[5], Hassan Mdala[6], Vera Bukachi[7]

[1] Global Geoscience, British Geological Survey, Keyworth, NG12 5GG, United Kingdom
[2] Department of Geography, King's College London, London, WC2B 4BG, United Kingdom
[3] Multi-Hazards and Resilience, British Geological Survey, Edinburgh, EH14 4AP, United Kingdom
[4] Geodynamik, Universität Tübingen, 72074 Tübingen, Germany
10 [5] Practical Action Consulting, Rugby, CV23 9QZ, United Kingdom
[6] Geological Survey of Malawi, Zomba, Malawi
[7] Kounkuey Design Initiative, Masera House, Kenyatta Market, PO Box 21972-00505, Nairobi, Kenya

15 *Correspondence to*: Joel C. Gill (joell@bgs.ac.uk)

**Abstract.** Reducing disaster risk is critical to securing the ambitions of the Sustainable Development Goals (SDGs), and natural hazard scientists make a key contribution to achieving this aim. Understanding Earth processes and dynamics underpins hazard analysis, which (alongside analysis of other disaster risk drivers) informs the actions required to manage and reduce disaster risk. Here we suggest how natural hazard research scientists can better contribute to the planning and 20 development of sustainable and resilient communities through improved engagement in disaster risk reduction (DRR). Building on existing good practice, this perspective piece aims to provoke discussion in the natural hazard science community about how we can strengthen our engagement in DRR. We set out seven recommendations for enhancing the integration of natural hazard science into DRR: (i) characterise multi-hazard environments, (ii) prioritise effective, positive, long-term partnerships, (iii) understand and listen to your stakeholders, (iv) embed cultural understanding into natural 25 hazards research, (v) ensure improved and equitable access to hazards information, (vi) champion people-centred DRR (leaving no one behind), and (vii) improve links between DRR and sustainable development. We then proceed to synthesise key actions that natural hazards scientists and research funders should consider taking to improve education, training, and research design, and to strengthen institutional, financial and policy actions. We suggest that these actions should help to strengthen the effective application of natural hazards science to reduce disaster risk. By recognising and taking steps to 30 address the issues raised in these recommendations, we propose that the natural hazard science community can more effectively contribute to the inter/transdisciplinary, integrated work required to improve DRR.





## 1 Introduction

This paper considers how natural hazard research scientists can better contribute to the planning and development of sustainable and resilient communities through improved engagement in disaster risk reduction (DRR). We target natural hazard scientists with an interest in contributing to sustainable development and resilience building, but who are uncertain of what steps to take. Collectively we as authors represent organisations in academia, the public sector and civil society with expertise from a range of countries and hazard settings. We reflect on existing good practice and identify how the natural hazard science community (including geologists, seismologists, volcanologists, hydrologists, meteorologists) can strengthen the translation, adoption and effective application of their understanding of physical processes and hazards to reduce disaster risk. While recognising the many debates relating to terminology, in order to inform the reader, we set out in **Table 1** key terms and definitions used throughout this paper.

Natural hazards (e.g., landslides, earthquakes, floods) have a significant impact on lives, livelihoods and economic growth, disproportionately affecting the most vulnerable in society and threatening development progress (Pelling et al., 2004). Between 1998 and 2017, disasters resulted in direct economic losses of US$2,908 billion, 1.3 million fatalities, and 4.4 billion people injured, rendered homeless, displaced or needing emergency assistance (CRED/UNDRR, 2018). To achieve the UN Sustainable Development Goals (SDGs), we must accelerate efforts to reduce impacts and diverge from a 'business as usual' approach (Spangenberg, 2016). The UN Sendai Framework for Disaster Risk Reduction (or 'Sendai Framework') aims to address this challenge, setting out a strategy to improve DRR (UNDRR, 2015).

While the Sendai Framework has a clear role for the natural hazard science community (Gill and Bullough, 2017), disasters are a complex and interdisciplinary challenge. Natural hazard scientists alone cannot provide the solutions necessary to ensure sustainable and resilient communities. The spatial and temporal occurrence of hazardous phenomena with exposure and vulnerability (both defined in **Table 1**) results in the generation of risk and potential for devastating effects. In this context, development challenges such as poverty, inequality, lack of access to, and overconsumption of, resources, climate change, and uncontrolled urbanisation can all drive changes to exposure and/or vulnerability, thus contributing to disaster risk (Pelling et al., 2004). Sustainable solutions require coherent engagement with diverse sectors and disciplines, including but not limited to the natural sciences. From our observations of research processes and collaborations within and beyond the international natural science community, we recognise some emerging trends, including:

● More *interdisciplinary research* between the geosciences and the social sciences (Schlosser and Pfirman, 2012; Van Noorden, 2015; UKCDS, 2016; Stewart and Gill, 2017), such as that described in Hicks et al. (2014), Martinez et al. (2018) or Barclay et al. (2019).

● Increased emphasis on *international, cross-sectoral partnerships* (Carabine et al., 2015; UKCDS, 2016; Dodson, 2017), such as those facilitated by the UK Global Challenges Research Fund (UK Government, 2020).





These trends are positive and offer opportunities for natural scientists to enrich their research, embed it into policy and practice, and help deliver development impact.

65   Building on existing good practice, this perspective piece therefore aims to provoke discussion in the natural hazard science community about how we can make the most of these opportunities and strengthen our engagement in DRR. In **Section 2**, we set out seven recommendations for improving the integration of natural hazard science into DRR. In **Section 3**, we synthesise key actions that natural hazards scientists and research funders can take to improve education and training, research design and methods, and partnerships and practice. In **Section 4**, we summarise some of the key benefits to the

70   natural hazard community and conclude that by taking specific steps, the natural hazard community can better contribute to interdisciplinary, integrated work to improve DRR. We acknowledge that not all natural hazard scientists need to work across all the proposed areas and that there is a clear requirement for disciplinary specialism. However, it is critical for natural hazard scientists to be aware of the broader DRR landscape, and opportunities for co-benefits to both the natural hazards community and society through enhanced ways of working.

75   By recognising and taking steps to address the issues raised in these recommendations, we propose in **Section 4** that the natural hazard science community can more effectively contribute to the inter-/transdisciplinary, integrated work needed to improve DRR.

## 2 Seven Recommendations to Ensure Natural Hazards Science Supports Effective Disaster Risk Reduction

### 2.1 Characterise (Multi-)Hazard Environments

80   Understanding disaster risk, the first *Priority for Action* within the Sendai Framework, includes the need to understand hazard characteristics and the natural environment (UNDRR, 2015). Ongoing geoscience research into surface and subsurface processes and the resultant formation of natural hazards remains essential. To better support DRR, however, we should consider in a comprehensive and systematic manner the range of hazard types, multi-hazard relationships and hazard scales that could occur in any given region, and how this hazard landscape may change over time. Many communities

85   around the world are exposed to multiple natural hazards, which do not always occur independently (Kappes et al., 2012). Relationships between hazards may exist that generate chains or networks of hazards (Gill and Malamud, 2014; Duncan et al., 2016; AghaKouchak et al., 2018). Understanding the 'multi-hazard' landscape of a region gives a better understanding of risk, and can help to inform management priorities, ensuring actions taken to reduce vulnerability to one hazard do not inadvertently increase vulnerability to others (Tobin and Montz, 1997; ARMONIA, 2007; Kappes et al., 2010; Gill et al.,

90   2020). Whilst work is being undertaken towards this objective, a single-hazard approach to research and dissemination is still dominant (Ciurean et al., 2018). This can result in technical excellence with respect to single-hazard research but hinders cross-disciplinary learning and reduces multi-hazard dialogue.



Literature describing approaches to understand multi-hazard relationships is limited, often focused on simulated environments, and combinations of two hazards, rather than methods examining real multi-hazard environments exposed to interrelating hazards (Ciurean et al., 2018). Understanding multi-hazard risk requires new approaches to knowledge infrastructures (i.e., the networks of people, institutions and processes concerned with the world's knowledge), data collection and management, database structure and hazard modelling to understand case histories and potential future scenarios of risk. For example, databases that record losses from disasters could be adapted to reflect the multi-hazard nature of the hazards involved and improve attribution of disaster losses to specific processes within this multi-hazard disaster (e.g., Froude and Petley, 2018). In terms of training and organisational management, we propose that more space (e.g., physical office space, space on a curriculum) should be dedicated to working across disciplines and identifying the connections between single hazards. Hemingway and Gunawan (2018) and Golnaraghi (2012) outline principles and successful examples of multi-hazard partnerships at the national level.

New approaches to data collection are needed to better characterise multi-hazard environments. These include consideration of different scale events, ensuring that low magnitude, frequent events are considered. Smaller magnitude events and their impacts are often not recorded because they are below the resolution of recording methods (Guzzetti et al., 2012) or do not qualify as an 'event' due to an imposed threshold (Gall et al., 2009). Yet, particularly in the Global South (so called 'developing countries'), the cumulative impact of these small, frequent hazards (also known as 'extensive hazards', UNDRR (2009)) can outweigh the impact of larger events, as well as erode the coping capacity of communities when high magnitude events do occur (Bull-Kamanga et al., 2003). The integration of data from diverse sources (e.g., fieldwork, published literature, grey literature, interviews to capture local perceptions of hazards, and questionnaires) can help to understand more fully the hazard environment. Examples include the DesInventar database, which primarily collects records from local newspaper archives to investigate events where only a small number of people were affected (Satterthwaite et al., 2018). Compiling detailed databases is time consuming but provides a more complete body of evidence to understand the full characteristics of hazards affecting a region, and a more accurate spatial pattern of mortality and morbidity (Osuteye et al., 2017).

The hazard environment is not static but can change due to natural forcing or anthropogenic activity, including climate change. Such processes can change the likelihood of natural hazards occurring, as well as hazards triggering or catalysing other hazards (Gill and Malamud, 2017; AghaKouchack et al., 2018). For example, road construction can increase the likelihood of landslides being triggered during an earthquake or heavy rain (Montgomery, 1994; Owen et al., 2008). Long-term studies of dynamic landscape changes due to anthropogenic activity are often beyond the lifecycle of research projects. Such studies may require different ways of working, such as establishing partnerships with organisations with a long-term presence in an area.



**Suggested Actions/Priorities for Change:**

- In both training and operational settings, space should be dedicated to working across disciplines to identify a fuller range of hazards and their potential interactions (or coincidence in time).

- New ways to collect data on, and analysis of, multi-hazards are needed, progressing from the consideration of two hazard types in simulated scenarios to multiple hazard types in real-world contexts (Ciurean et al., 2018).

- Enhanced communication across disciplines can help to facilitate dialogue relating to risk from 'multi-hazards'. We
encourage leadership from geoscience unions, research funders, and professional associations to facilitate more cross-hazard cooperation through joint meetings and collaborative working spaces.

## 2.2 Prioritise Effective, Positive, Long-Term Partnerships

Reducing disaster risk requires generating and utilising knowledge from across disciplines and sectors (UNDRR, 2015; Twigg, 2015). Recognition of the complexity of risk has led to an increase in calls for and application of interdisciplinary
partnerships to disaster risk/resilience research, integrating natural and physical science knowledge, methods and/or approaches with the social sciences, arts and humanities. The drive to link research to practice and the participation of 'non-scientists/specialists' in the design and implementation of disaster risk science (transdisciplinary research practice; Horlick-Jones and Sime, 2004; Hilhorst and Heijmans, 2012), calls for strong partnerships. This acknowledges the need to work with those at risk rather than viewing them as research 'subjects' or 'recipients' (Pelling, 2007).

Partnerships, emphasised in the Sendai Framework targets and guiding principles, are key to harnessing knowledge, to better understand and address the problems faced by those at risk (see **Section 2.3**). Partnerships can be of many different kinds (including networks and collaborations) and when established effectively, they can increase the impact of DRR initiatives by ensuring their sustainability, replicability, and better use of resources (Twigg, 2015). Partnerships can be both vertical (global, regional, national to local) and horizontal (across sectors and disciplines) (Twigg, 2015). Existing connections are
often the best starting point; the Sendai Framework recommends that science contributions to DRR can be enhanced through the coordination of existing networks and scientific institutions at all levels and regions (UNDRR, 2015). Networks can create an enabling environment for knowledge sharing, development, and technology transfer (Sakic-Trogrlic et al., 2017). For example, the Global Volcano Model network successfully coordinated the input of >130 scientists for the first review of volcanic hazards and threats in the 2015 Global Assessment Report (Loughlin et al., 2015). The ongoing UK Global
Challenges Research Fund (GCRF) project, Tomorrow's Cities, aims to enhance risk-sensitive urban development through a global network of integrated research programmes, led by local teams in low-to-middle income countries (Tomorrow's Cities, 2020).

Although essential, partnerships can be difficult to establish and maintain. They take time, negotiation, sustained effort, transparency, trust, resources, commitment and institutional support (Twigg, 2015). Ensuring researchers understand their





role in DRR policy and practice, and likewise the role and responsibilities of partner institutions, underpins effective, equitable and trusting partnerships. Similarly, recognising the distinction between those institutions with operational mandates and those undertaking research is critical to ensuring that research supports, rather than undermines, national and local capacity (see Newhall et al., 1999).

The ELHRA Guide to Constructing Effective Partnerships (ELRHA, 2012) provides a useful overview of the benefits and
challenges of collaborations between humanitarian and academic organisations, and provides practical guidance on identifying, establishing and maintaining effective partnerships. Generic guidance on how individual academics and organisations build more effective partnerships with, for instance, national science institutions outside of their own country is not common. An initial step would be to see whether any internal policies exist to support this, such as guidance for working overseas (e.g., obtaining research permissions), data management policies (e.g., data sharing and Intellectual Property
Rights), and ethics policies and frameworks. The co-establishment of Memoranda of Understanding, including codes of practice and ethics, can form the basis for effective, lasting institutional partnerships, they are underpinned with funding to support the individuals sustaining these partnerships. Documenting effective partnership examples (and any challenges) and sharing these with the wider research community would also benefit those researchers new to building networks and collaborations.

**Suggested Actions/Priorities for Change:**

- Higher education and ongoing professional development training should include partnership development topics (e.g., project management, facilitation skills, and inter- and transdisciplinary working).

- Natural hazard scientists should implement ethical frameworks for building and maintaining equitable partnerships (see Conway and Waage, 2010).

- Funding opportunities should recognise, and provide for, the time and resource required to build partnerships (e.g., attend in-person meetings or conferences). Where remote working is required, virtual communication tools, social media and fora such as groups on ResearchGate (2020) can initiate dialogue. Consideration should be given to who might be missing from these partnerships and how they could be engaged through the project.

- Institutional support for partnerships, through Memoranda of Understanding between institutions, for instance, can
ensure mutually agreed expectations, codes of practice and ethics. Roles and responsibilities within partnerships should be discussed and clarified. One project management approach is to use a RACI diagram (see https://pmdprostarter.org/raci-diagram/), capturing information on key responsibilities, accountabilities, who should be consulted, and who should be informed.

- Funding for researchers based in the region of study to help strengthen both knowledge exchange and sustainability of
the impact.



## 2.3 Understand and Listen to your Stakeholders

Understanding the priorities, interests, ambitions and challenges of stakeholders is essential to developing and undertaking effective DRR research. Stakeholders might include different researchers across many disciplines, government agencies, non-governmental agencies, civil society, the private sector, and communities at risk (Twigg, 2015). Stakeholder mapping (identifying and understanding stakeholders in a given project, how they sit within and influence a system) is an important task when connecting natural hazards science to DRR, although a significant undertaking. There are many tools to support this process, with one illustrated in **Figure 1**. Stakeholder mapping identifies the relative interest and influence of stakeholders throughout the project lifecycle, thereby allowing the project team to focus and adapt their engagement during the project. Stakeholder mapping may help identify potential partners (**Section 2.2**), or existing partners may assist in identifying who the stakeholders are.

Consultation with stakeholders should help inform the types of research activities undertaken. There are numerous examples where a hypothesis, or a proposed solution, transpires not to be a stakeholder priority, and thus research outputs struggle to gain traction (Clot, 2014; Schipper and Pelling, 2006). The Sendai Framework advocates for opportunities for Global South nations to identify and express their needs and priorities, and for countries in the Global North (so called 'developed countries') to actively listen to them (UNDRR, 2015). Asking stakeholders 'what is important to you?' can open up new ideas and avenues of effective collaboration. In Malawi, Leck et al. (2018) highlight how international NGOs had set the agenda for local DRR, undermining the authority of local governments. To solve problems that are both scientifically novel and societally relevant, it is key to have awareness of local context (political, economic, social, cultural, technological, legal, and environmental) and local perceptions of risk through meaningful engagement and co-production of research with stakeholders.

Rather than starting research with systematic data collection to test a hypothesis, time may be better devoted to the development of a research question that is informed by stakeholder needs and an understanding of local context. For example, instead of beginning with the aim of developing a GIS model for visualising earthquake risk, more effective research may commence by investigating who manages earthquake risk, how this is done, what scientific and organisational challenges they face (e.g., which stakeholders are involved, what other priorities do they have, and what other constraints exist on their time), and what their technical, time and scientific capacity is to develop and adopt new ways of working. Ideally, the project aims and hypothesis would then be developed in collaboration with stakeholders. Ensuring partners from the Global South (e.g., researchers, NGOs) are co-investigators on research proposals can bring contextual understanding into project design and implementation. Ongoing conversations and assessment of prototypes may be needed to help stakeholders articulate their needs and help researchers understand what research and methods are relevant, over time and



through careful management of this important relationship. This dialogue and the emerging contextual understanding can guide more effective hypothesis development and data collection.

'Theory of Change' (Weiss, 1995) is one approach that could enable this co-production, with resources available online (e.g.,
DIY Toolkit, see references). Theory of Change starts by using a context analysis to identify the problem to be solved (e.g., reducing deaths from tsunamis), and then works backwards to characterise root drivers of this problem (e.g., ineffective early warning systems), the key audiences for implementing change (e.g., civil protection and community groups), the access points and motivators for those groups, steps required to bring about change, and the broader benefits. At each stage, the Theory of Change approach gives attention to uncertainties and assumptions. Practitioners, donors and academics have
applied Theory of Change effectively in different ways (Vogel, 2012), helping to identify key stakeholders and create a roadmap to achieve real change through applied research. While time consuming and often challenging to follow within the typical cycle of funding applications, this approach can be fundamental to developing research programmes that result in improved DRR.

**Suggested Actions / Priorities for Change:**

● Higher education and ongoing professional development training should cover stakeholder mapping, managing people in projects, facilitation skills (including running workshops), and transdisciplinary working.

● Training for natural hazard scientists on how to ethically identify stakeholders and co-produce research questions using techniques such as Theory of Change.

● Developing long-term relationships with applied partners such as NGOs and national institutions (e.g., geological
surveys, hazard monitoring agencies) who have a long-term presence in and access to a range of stakeholders.

● Mechanisms (e.g., funding for time and networking) to include non-academic partners and stakeholders in research proposals, co-develop transdisciplinary research questions, identify desired outputs, and understand stakeholder capacities.

**2.4 Embed Cultural Understanding into Natural Hazards Research**

Culture, defined in **Table 1** (but we recognise other definitions exist), includes the social institutions, customs and beliefs that people hold, as well as the characteristics that unite people (Cannon and Schipper, 2014). Examples include religious beliefs, traditional beliefs, values, livelihood choices, settlement patterns (Canon and Schipper, 2014). We are part of and affected by culture as researchers, and this can shape the way in which we approach ideas or partnerships as natural hazard scientists. Culture can also affect risk, by either increasing or reducing the vulnerability of individuals and communities,
shaping the norms by which the acceptability of risk is defined, and influencing how people respond to and cope with disasters (Bankoff, 2003; Schipper and Dekens, 2009; Canon and Schipper, 2014; O'Connell et al., 2017). Examples include:



- Indigenous knowledge and culture is attributed to the very high survival rate following the 26 November 1999 tsunami on Pentecost Island, Vanuatu (Walshe and Nunn, 2012).

- The cultural expectation that women are caregivers was shown to increase the physical exposure of women to illness
and the psychological burden post disaster in Manila, Philippines (Reyes and Lu, 2016).

- Local sub-cultures at Merapi volcano, Indonesia, were found to influence local community actions during frequent eruptions (Donovan et al., 2012).

Understanding culture is therefore important when considering how to reduce disaster risk. The Sendai Framework notes that DRR policy and practice should integrate cultural perspectives and advocates for the creation of 'cultures of prevention' (vs.
response) and maintenance (vs. disrepair which can increase physical vulnerability) to be established (UNDRR, 2015). Therefore, natural hazard scientists not only should understand how culture relates to disaster risk response and reduction, but may also need to work with experts in behavioural science to help drive changes in established cultures. Understanding culture is a critical part of the context analysis described in **Section 2.3**, done before research, and will require the strong partnerships advocated for in **Section 2.2**. While ethnographic research (i.e., immersion in a group for an extended period,
observing behaviour, listening to what is being said and asking questions, Bryman, 2016) to understand people and their cultures is not part of natural hazard science training, the outcomes of such research could enhance the work of natural hazard scientists and help to maximise the impact. Examples include:

- *Enriching Data:* Our understanding of historical occurrences of natural hazards informs our characterisation of the potential for future events and their likely magnitudes. Natural hazard science has traditionally understood past events
through historical archives, instrumental records, and field observations. Understanding culture is critical to identifying how information is better captured and communicated in any given location. Stories passed down from one generation to another, for example, may be a significant record of information and help to enrich data collected using traditional fieldwork and in locations where written and instrumental records are minimal (Cronin and Kashman, 2008).

- *Contextual Understanding:* Understanding cultural beliefs, practices and rituals can also help researchers to be sensitive
to the associations people have with hazardous areas (e.g., the religious significance of a volcano), as well as understand their coping capacities and resilience of communities living in hazardous areas (e.g., Cronin et al. 2004).

- *Improving Research Dissemination:* Many grant applications require participants to outline research impact on society, and how information will be disseminated to stakeholders. Dissemination should be done in a way that is acceptable and understandable to stakeholders (**Section 2.3**), which will vary (e.g., regionally). Understanding culture can help to guide
decisions about the appropriate nature of research outputs (e.g., storytelling, radio shows, briefing notes, films, theatre (e.g., Hicks et al., 2017), and who is best placed to share these.



Natural hazard scientists sit within their own cultures, and this positionality (defined in **Table 1**) is likely to affect their approach to research, and interactions with others. Researcher positionality could be integrated into the training of hazard scientists. For example, before a researcher engages in work in an unfamiliar or different cultural context to their own, they

should reflect on how their experiences, values and beliefs could influence or prejudice whom they may consult, the questions they may ask, data they gather, 'products' they advocate for, and appropriate conduct. Individual perspectives on religion, for example, may mean a researcher is reluctant to collaborate with leaders of faith-based organisations. Analyses of the 2014–16 Ebola crisis in Sierra Leone, however, demonstrates that faith leaders can play a transformational role in communicating key humanitarian messages (Featherstone, 2015). Likewise, positionality includes consideration of the

assumptions that might be extended to researchers by stakeholders and participants, particularly around issues of trust and equality. Being aware of the implications, for instance, of being a researcher from a high-income country working in a low-income country, in terms stakeholder expectations is critical.

**Suggested Actions/Priorities for Change:**

- When developing research partnerships, natural hazard scientists should consider including those with ethnographic
290        training (e.g., geographers, historians, anthropologists), or identify existing and relevant ethnographic knowledge in publications and reports.

- When planning research dissemination strategies, public outreach, and hazards education initiatives, in addition to their partners and stakeholders, natural hazard scientists could consult literature, historians, anthropologists to understand cultural constraints, challenges and opportunities.

- Train natural hazard scientists to understand and reflect on their own positionality, providing them with the skills to understand how their belief systems may influence their work.

### 2.5 Ensure Improved and Equitable Access to Hazards Information

Hazard information should reach those in need, be understood, and be acted on if it is to help reduce risk (Mohadjer et al., 2016). This requires the gap between knowledge generation and knowledge access to be addressed (Aitsi-Selmi et al., 2016).

It is often those most vulnerable to the impacts of disasters who struggle to access useable hazards information. The natural hazards science community should consider not only equitable access, but also how to ensure that all stakeholders can act on hazards information. As demonstrated in **Sections 2.2**–**2.4**, working in partnership, listening to stakeholders and culturally contextualising research can help to create useful hazard information. **Table 2** further outlines the many factors that can enhance access to hazard information.

Natural hazard scientists should be aware of and sensitive to any barriers if they are to deliver information in an appropriate form, a timely manner, and a way that facilitates action by stakeholders (Scienseed, 2016). Training for hazard scientists could draw on good communications practice to strengthen their ability to make natural hazards science more accessible to



groups outside of the professional community. Consideration should be given to the audience and their needs, including (amongst other characteristics) their values, attitudes, concerns, knowledge, language, and personal and social aspirations
(Liverman, 2008). Scientists who wish to inform decision making should use this understanding to tailor information to their audience's specific needs. Useful hazards information also takes into account the technical limitations of data. For example, the most common assessment methods for seismic hazard are Probabilistic Seismic Hazard Analysis (PSHA) and Deterministic Seismic Hazard Analysis (DSHA). Though useful for developing building codes, PSHA may be misleading in locations where data are sparse (Stein et al., 2018), and other methods may be required (Robinson et al., 2018).

Natural hazard scientists should therefore work collaboratively with partners and stakeholders to develop hazard information products with their intended audience. The most effective method of understanding informational needs of stakeholders is to establish and nurture a two-way communication, co-production, between scientists and decision-makers, building relationships, trust, and credibility over time (Morss et al., 2005). This dialogue will help to guide the choice of language and content of hazard information products to make them more appropriate for stakeholders. This includes the spoken language,
but also the terminology and level of understanding the content is pitched at stakeholders.

**Suggested Actions/Priorities for Change:**

● Natural hazard scientists should pursue open-access publishing, and/or write short, accessible summaries of their research (e.g., policy briefs) to be disseminated to appropriate stakeholders.

● Good communication practice should be essential training for natural hazard scientists, exploring the importance of
understanding and tailoring information to specific audiences, and co-developing hazard information products with intended audiences.

● Working with partners and stakeholders (co-production) is key to the creation of useable hazards information.

### 2.6 Champion People-Centred DRR - Leaving No-One Behind

The SDGs and Sendai Framework both emphasise 'leaving no-one behind' and ensuring that the poorest and most
vulnerable in society have access to the resources, information, and support required to effectively reduce risk and encourage sustainable development. People's ability to prepare for, respond to and recover from disasters is shaped by an array of social, cultural, economic and political factors (Wisner et al., 2012). Vulnerability to hazards is exacerbated by existing social stigmatisation and isolation, and those who are marginalised in society are often the most vulnerable in facing natural hazards (Pincha, 2008; Wisner et al., 2012; Gorman-Murray, 2017). For natural hazard scientists, this means acknowledging
that risk reduction is not simply about the hazard, but also the analysis and understanding of vulnerability (often the weaker component of risk analysis (Schneiderbauer and Ehrlich, 2006)) and actively reflecting upon where we work, with whom we work, and how we work. This may involve consideration of our own positionality (outlined in **Section 2.4**) in terms of how we understand marginalised groups. It also requires informed and difficult decisions, balancing the choice to work in areas




where some marginalised groups are located (e.g., fragile states, regions with active conflict, and regions where
humanitarian workers are threatened), against whether and how natural hazard scientists can safely, ethically, and effectively
work in these regions.

Marginalised groups risk being excluded from all aspects of DRR, including understanding hazards and risk. In a study of
flood management in Jakarta, van Voorst and Hellman (2015) found that strategies to increase rainfall infiltration in open
spaces had been ineffective due to these spaces being occupied by marginalised groups who did not appear on the official
city map. This example highlights how the uncertainties and politics of information used in a seemingly 'neutral' hazard
assessment may have unanticipated outcomes. Proactive effort is recommended to reach out to, collaborate with, and listen
to the voices of marginalised groups, with careful consideration of which voices are missing (Brown et al., 2019), starting
with stakeholder identification (**Section 2.3**). Marginalised groups may be more vulnerable to disasters, but they also have
valuable knowledge, skills, experiences and coping methods that should not be overlooked or ignored.

Leaving no one behind also means better engagement with indigenous communities and integration of local and indigenous
knowledge and perceptions into disaster risk reduction. Environmental history, passed between generations through
storytelling, can be an important source of information (see **Section 2.4**) enriching the data used to understand the multi-
hazard landscape of a region (see **Section 2.1**). Approaching the topic of local knowledge requires designing fully
participatory approaches to reflect its heterogeneity, both in terms of content and distribution within the community (Sakic-
Trogrlic et al., 2019). Mercer et al. (2010) set out a framework to integrate indigenous and scientific knowledge for disaster
risk reduction. Such approaches, and an exploration of their strengths and criticisms, are not typically included in the
curricula of subjects training natural hazard scientists (e.g., Earth science). This may hinder the extent to which natural
hazard scientists accept the validity of local and indigenous knowledge, proactively engage with this as a source of evidence,
and integrate it into hazard assessments. Other groups at risk of being left behind are children and youth, with themes
relating to natural hazards and disaster risk often not included in the school curricula for those in the Global South.

**Suggested Actions/Priorities for Change:**

- Increase reflection on *how* natural hazard scientists ensure meaningful participation in research and outreach activities by underrepresented, vulnerable and marginalised groups.

- Include training on integrating local and indigenous knowledge into natural hazard assessments and disaster risk
reduction.

- Introduction of natural hazard and DRR related topics in the curriculum at lower education levels.

**2.7 Improve Links between DRR and Sustainable Development**

DRR can drive forward and protect development progress and is therefore embedded within 10 of the 17 SDGs. Goals on
poverty, hunger, health, education, water and sanitation, infrastructure, cities, climate change, oceans and terrestrial





ecosystems all refer to risk reduction, building resilience, early warning, or adaptation (United Nations, 2015). Furthermore, effective sustainable development interventions (e.g., addressing inequalities, increasing access to resources, better planned urbanisation) can increase individual, community, institutional and infrastructure resilience by reducing exposure and/or vulnerability (Pelling et al., 2004). Examples of both relationships include:

- *SDG 11 (Sustainable Cities).* Embedding understanding of the subsurface (e.g., geotechnical properties, shallow
geohazards potential) into urban planning can increase the safety of urban development (Mielby et al., 2017).

- *SDG 4 (Quality Education).* Increasing access to education can reduce vulnerability to natural hazards by increasing understanding of Earth dynamics and environmental change, and exploring steps to reduce risk (Mohadjer et al., 2018).

While there is a growing awareness of the relationship between DRR and sustainable development, it is not yet clear whether this is embedded within the natural hazards community. Gill and Bullough (2017) noted that only 19 of 1059 sessions at the
2017 European Geoscience Union (EGU) General Assembly referred to the SDGs, Sendai Framework or Paris Agreement. Furthermore, of the 1268 abstracts submitted to sessions within the Natural Hazards Division of the 2019 EGU General Assembly, only two referred to 'sustainable development'. Based on a Google Scholar search (18 November 2019), of 697 articles published in the EGU journal Natural Hazards and Earth System Sciences between 2015 and 2018, 35 (5%) make direct reference to sustainable development. Whilst acknowledging that some studies may be contributing to sustainable
development through other research outputs, what these statistics suggest is that some natural hazard scientists may be missing opportunities to address research questions of local/national priority expressed through relevant development strategies (e.g., Kenya Vision 2030). This is increasingly a demand made by research funders, for example, it is embedded within the UK Global Challenges Research Fund strategy (UK Government, 2020). Aligning research power with sustainable development ambitions expressed in these strategies can help to secure critical 'pathways to impact' that help to
embed natural hazards research into risk reduction and embed risk reduction into sustainable development.

Translating natural hazards science into tools that support sustainable development policy and practice requires sustained and effective dialogue (Lubchenco et al., 2015). This may require new partnerships (see **Section 2.2**) and communication methods (Marker, 2016; Stewart and Gill, 2017), to strengthen coherence between different policies, to mainstream DRR and avoid a policy in one sector increasing vulnerability to natural hazards. The importance of policy coherence is embedded
within the SDGs (United Nations, 2015), and articulated as being critical to climate change adaptation (England et al., 2018).

**Suggested Actions/Priorities for Change:**

- Increase awareness of how individual natural hazards research projects join-up and relate to regional, national and local sustainable development, disaster risk reduction and disaster risk management strategies.

- Embed training in public policy into natural hazards science courses at university level.



## 3 Discussion and Cross-Cutting Themes

In **Section 2**, we reflected on seven ways those working on natural hazards science can enhance their contribution to DRR, integrating examples of good practice and innovative solutions where appropriate. In **Table 3**, we synthesise priorities for change proposed in **Sections 2.1** to **2.7**, grouping these into changes linked to (i) education, training and continued professional development, (ii) research priorities, methods and approaches, and (iii) institutional, financial and policy actions. Each of these would benefit from aligned funding.

Whilst the seven recommendations could be conceived as utopian, we have provided some practical steps that build on the existing skills and strengths of natural hazard scientists. In addition, we have identified where enablers, such as training programmes and funding, are required. Action to achieve one of the recommendations in **Section 2** (e.g., prioritising positive partnerships, **Section 2.2**) could also reinforce other changes (e.g., ensuring equitable access to appropriate information, **Section 2.5**). Although we set out seven distinct themes in **Section 2**, we recognise there are interactions and note the importance of thinking across these themes in an integrated manner. For example, a professional skills module in undergraduate or postgraduate courses that integrates communication, policy engagement, stakeholder mapping and partnership development training could help deliver many of the ambitions expressed in **Table 3**.

The vision of change we present requires transformation to natural hazard science education and training, introducing new skills and exposing scientists to a wider range of disciplinary knowledge, along with the option to learn interdisciplinary and transdisciplinary research approaches. This includes recognising the role of local or indigenous knowledge, demonstrated to be key to community-level risk reduction (e.g., Sakic-Trogrlic et al., 2019). Cultural and ethical understanding, cross-disciplinary communication, and social science research approaches can enhance our science but are not typically included in a natural hazard scientist's training (Lubchenco et al., 2015; Gill, 2017; Stewart and Gill, 2017). Effective communication is a repeated theme in many of the recommendations in **Section 2**, across sectors, disciplines and cultures. Yet much of the existing communication training offered to university students focuses on communicating natural hazards science to fellow natural hazard scientists or the public who reside in the same national context as the place they are a student. In contrast, it could be enriching to bring students together from geoscience, engineering, anthropology, health sciences, geography and the political sciences to explore their research tools, information requirements, and preferred ways of giving and receiving information. Cross-disciplinary engagement at an early stage of a career would likely result in a strengthened understanding of ethics, and appreciation of interdisciplinary partnerships throughout their work.

Reforms to the training of natural hazard scientists, should be complemented by the adoption of different approaches to determining research questions, building research partnerships, and connecting research to decision makers. Effective partnerships, with clear roles and responsibilities are important (Sargeant et al., 2018), and these will increasingly include a wider variety of skills and disciplines (e.g., ethnographers, behavioural scientists). Equitable and ethical partnerships take time to develop and maintain, but this should not be an excuse for poor partnership practice. Natural hazards scientists in the



Global North have a professional responsibility to listen to the needs and priorities of natural hazard scientists and stakeholders in Global South nations, and work with them to address these. Working in partnership, and listening to stakeholders, is fundamental to understanding critical aspects of local context, ensuring effective communication, and
including marginalised groups. This process of listening is not a one-off exercise, but iterative and requiring continual engagement.

Whilst individual behaviours can promote change, we recognise that there are number of institutional and financial transformations required, including improving funding mechanisms to include non-academic partners in research proposals, supporting the development of new training schemes and providing funding for open-access publishing.

**4 Conclusions**

This perspective paper has provided evidence and recommendations for how natural hazard scientists can contribute to reducing disaster risk, and securing the ambitions of the SDGs. Natural hazard scientists' understanding of Earth processes and dynamics underpins hazard analysis, which (alongside analysis of other disaster risk drivers) in turn informs the actions required to manage and reduce disaster risk. This paper recommends actions the natural hazard science community can take
to enhance the contribution of their work to the planning and development of sustainable and resilient communities. We recommend changes to (i) education, training and continued professional development, (ii) research design, methods and implementation, and (iii) institutional, financial and policy actions, to strengthen the translation, adoption and effective application of their understanding of physical processes and hazards to reduce disaster risk. In addressing the priorities for change set out in **Section 2**, and summarised in **Table 3**, we propose the following benefits:

• *Richer Data and Better Understanding of the Physical Multi-Hazard Landscape.* Improved integration of data characterising different natural hazards, from a wider range of sources (e.g., integrating indigenous knowledge), will enable a richer understanding of the multi-hazard landscape and potential complex and compound hazard scenarios.

   • *Improved Capacity Building.* Natural hazard scientists will grow in their awareness of capacity (of individuals and organisations), and their ability to develop the capacity of others through locally- and culturally- appropriate means.

• *Better Partnerships.* Natural hazard scientists will work in a more ethical manner, with greater sensitivity of context to support (vs. undermine) other partners. Natural hazard scientists will have a clearer understanding of how to engage with vulnerable communities, increase their access to information, and actively reflect on if and how they are able to participate in DRR activities.

   • *Increased Access and Use of Natural Hazards Science.* Listening to stakeholders' questions, understanding their
decision-making processes, and building cultural understanding can inform the natural hazards science that is done and the way this is shared with others to encourage the embedding of science within policy and practice.

Together these benefits will support DRR, and the development of sustainable and resilient communities. It is now the responsibility of individual natural hazard scientists and those in positions of leadership (e.g., course directors, funding agencies) to consider how the recommendations set out here apply to their work, and what more they can do to ensure

natural hazards science helps realise the ambitions of the Sendai Framework and UN Sustainable Development Goals.

**Author Contribution**

All authors contributed to the intellectual discussion preceding this paper, determining the themes to focus on. JCG and FET led the writing of the paper, with significant input from MJD and SM. MB and HM contributed to the writing of the paper, with helpful comments provided throughout. VB reviewed the paper and contributed additional perspectives which enhanced

the final version.

**Competing Interests**

The authors have no competing interests.

**Acknowledgements**

The authors would like to acknowledge the colleagues, partners and stakeholders with whom which they have worked with
over their collective research and practitioner careers in DRR and sustainable development. These experiences have informed the authors' contributions to this paper. We thank Andres Payo Garcia and John Rees for thoughtful internal reviews prior to submission. JCG and MJD publish with the permission of the Executive Director, British Geological Survey (UKRI).

**Financial Support**

JCG and MJD contributions to this article were supported by British Geological Survey NC-ODA grant NE/R000069/1: *Geoscience for Sustainable Futures*, and Royal Society Challenge-Led Grant CHL-R1-180192: *Using a multi-hazard and catchment-based approach to understand and increase resilience in hyper-expanding cities in Vietnam and the Philippines*.

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



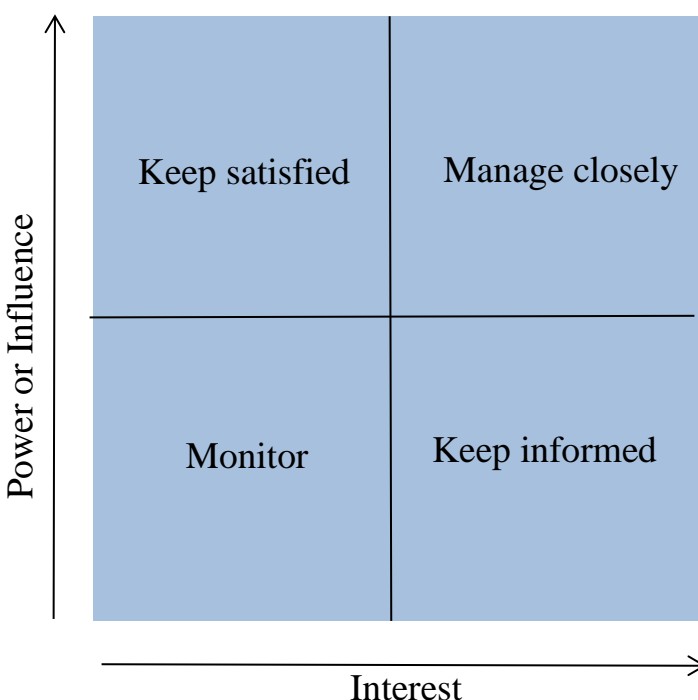


**Figure 1: Stakeholder Mapping Tool.** Determining the level of interest and influence (or power) of different stakeholders can help to ensure effective communication with each group. Figure adapted from Mendelow (1981).



**Table 1: Key terms and definitions used throughout this paper**

| Term | Definition | Source |
|---|---|---|
| Culture | The complex whole which includes knowledge, beliefs, arts, morals, laws, customs, and any other capabilities and habits acquired by [a human] as a member of society. | UNESCO, 2017 |
| Exposure | The situation of people, infrastructure, housing, production capacities and other tangible human assets located in hazard-prone areas. | UNDRR, 2017 |
| Interdisciplinary | Interdisciplinary studies address specific real world problems. This involves bringing people and ideas together from different disciplines (e.g., natural and social scientists) to collectively frame a problem, agree on a methodological approach and analyse data in an integrated manner. | Adapted from Hammer and Söderqvist, 2001 and Stock and Burton, 2011 (see references therein) |
| Partner | A "partner" is a person, organization, network or association who works collaboratively with others as part of a defined agreement, project or framework to achieve a common purpose or undertake a specific task and to share risks, responsibilities, resources, competences and benefits. | UNDRR, 2016 |
| Positionality | The stance or positioning of the researcher in relation to the social and political context of the study—the community, the organization or the participant group. | SAGE, 2014 |
| Resilience | The ability of a system, community or society exposed to hazards to resist, absorb, accommodate, adapt to, transform and recover from the effects of a hazard in a timely and efficient manner, including through the preservation and restoration of its essential basic structures and functions through risk management. | UNDRR, 2017 |
| Stakeholder | Any individual or group with an interest in reducing disaster risk (i.e., including those within a project, and external to but benefiting from a project). | UNDRR, 2016 |
| Transdisciplinary | Transdisciplinary studies go beyond interdisciplinary studies by placing emphasis on the participation of non-academic partners to solve real world problems, by differentiating and integrating knowledge from various scientific and societal bodies of knowledge. | Adapted from Stock and Burton, 2011 (see references therein) and Lang et al., 2012. |
| Vulnerability | The conditions determined by physical, social, economic and environmental factors or processes which increase the susceptibility of an individual, a community, assets or systems to the impacts of hazards. | UNDRR, 2017 |






**Table 2: Accessibility and Usability of Hazards Information**

| Theme | Challenges to Accessibility and Usability of Information | Actions to Improve Accessibility and Usability of Information. |
|---|---|---|
| **Communication Medium** | Differential access to and control over communication technology (the 'digital divide') is a barrier to reaching some marginalised groups, including women, in many parts of the world (GDN, 2009; Shrestha et al., 2014). | Multiple communication media to reach multiple vulnerable groups. |
| **Open-Access** | Many natural hazard reports, maps, databases and tools exist behind paywalls, with access limited to those who can pay. | Publishing in open access formats can make access, usage, and dissemination of data less time-consuming and resource-intensive (Mohadjer *et al.,* 2016). |
| **Language and Content** | Information may not be accessible if it is not appropriately tailored for an audience to understand and make decisions from. | Carefully explaining technical language, utilising diverse methods for communicating concepts such as uncertainty (e.g., Shepard *et al.*, 2018), publishing in appropriate local languages, considering literacy levels, and providing advice on specific actions to take to mitigate risks from natural hazards. |
| **Capacity of Hazard Scientists to Communicate** | Access to hazard information often depends on natural hazard scientists being proactive at disseminating beyond traditional scientific journals. | Additional training, beyond the scope of many traditional geoscience courses, to increase confidence in using different dissemination methods. |
| **Resource Availability** | Those who have access to information (e.g., hazard professionals) may not have the resources needed to share this information with those who do not have access (e.g., the general public). | Increased and better partnerships (**Section 2.2**), to help leverage the resources needed for effective communication. |
| **Timeliness of Information** | There may be a difference between when hazard information is needed, and when this can actually be generated (Robinson *et al.,* 2017). | Long-term, sustained partnerships (**Section 2.2**) can help to generate useful outputs rapidly by drawing on existing understanding of stakeholders, and their needs and capacities. |



**Table 3: Summary of Action Points to Help Improve Geoscience Engagement in DRR.**

| Recommendations | Suggested Actions | | |
|---|---|---|---|
| | **Education, Training and Continued Professional Development** | **Research Design, Methods and Implementation** | **Institutional, Financial and Policy Actions** |
| *Characterise (Multi-) Hazard Environments* | In training, space for enhancing communication and working across disciplines. | Improve methods to capture and document multi-hazard observations. Improve analysis of multi-hazard environments. | More cross-hazard cooperation through joint meetings and collaborative working spaces. |
| *Prioritise Positive Partnerships* | Include formal training in ethical and equitable partnership development. | Discuss and agree roles and responsibilities within partnerships. | Develop funding mechanisms to build and maintain long-term partnerships and include non-academic partners in research proposals. Increase opportunities for networking to facilitate partnership building (particularly for early-career scientists). Implement frameworks for ethical and equitable partnerships. |
| *Listen to and Understand Stakeholders* | Train natural hazard scientists in stakeholder mapping, co-production of research questions, and techniques such as Theory of Change. | Ensure research questions are driven by an understanding of local context, perceptions and stakeholder needs. | Develop long-term relationships with applied partners such as NGOs and the public sector. |
| *Embed Cultural Understanding into Natural Hazards Research* | Train natural hazard scientists to understand and reflect on their own positionality. | Broader research project teams and greater engagement with the ethnography community and/or literature. | Consult relevant expertise to understand cultural constraints, challenges and opportunities when planning research dissemination and hazards outreach and education initiatives. |
| *Ensure Improved and Equitable Access to Hazards Information* | Enhance communication training for natural hazard scientists. | Pursue open access publishing. Co-develop research outputs and dissemination. | Produce short, accessible summaries of hazards research (e.g., policy briefs) for stakeholders. |
| *Champion People-Centred DRR - Leaving No-One Behind* | Include training on integrating local and indigenous knowledge and perceptions into natural hazard assessments and disaster risk reduction. | Actively reflect on how underrepresented, vulnerable, and marginalised groups can meaningfully participate in and benefit from research. | Focus hazards education and outreach initiatives specifically on vulnerable and marginalised groups, including in schools. |
| *Improve Links between DRR and Sustainable Development* | Include training in public policy, to facilitate greater connection of hazards science to sustainable development priorities. | Consider how individual projects join-up and relate to regional, national, and local sustainable development, DRR and disaster risk management strategies. | |