# Peer review of "Invited Perspective: Building sustainable and resilient communities - Recommended actions for natural hazard scientists"

_Natural Hazards and Earth System Sciences, 2020_

## Referee Comment (RC1) · Anonymous Referee #1 · 24 Jul 2020

Gill and co-authors propose a very well-written and interesting perspective on recommended actions for scientists in the context of disaster risk reduction and the sustainable development goals. These recommendations are specifically addressed to researchers involved in the study of natural hazards with the aim that their engagement in DRR and sustainable development is strengthened.

What Gill and co-authors propose is timely and the proposed seven recommendations nicely encompass the whole context behind the study of natural hazards and the goal to achieve DRR. The authors have done a very good job of using a diverse literature that covers various natural hazards.

[Figure]

I have made few minor comments on the PDF that could improve the manuscript, which I will not repeat here. I have also pointed out few typos.

Please also note the supplement to this comment:
https://nhess.copernicus.org/preprints/nhess-2020-163/nhess-2020-163-RC1-supplement.pdf

**Supplement:**

[revised manuscript text omitted]

---

## Short Comment (SC1) · 6 Aug 2020

This is a great paper and makes some excellent points around the need for natural scientists to be trained in a much wider range of methods and approaches. One thing that I think is there but needs greater emphasis is the need for interdisciplinary working - you mention it multiple times, but there does not seem to be a lot of detail on the complexity of it or the expertise that is available from other disciplines to supplement that of natural scientists. It isn't necessary for natural scientists to become social scientists - the social sciences have a wide range of experience and expertise that is not very well respected a lot of the time in the natural/physical science community. This is

NOT just about "behavioural science" (which is a narrow field with very dubious epistemic foundations some of the time) - it also requires working with environmental social scientists (who use mixed methods), anthropologists and human geographers. A key requirement of such working is flexibility - social science typically involves different assumptions about what makes robust knowledge than the physical sciences do, and so awareness of and respect for that is of critical importance.

In summary - I'd just encourage the authors to think about this in a bit more detail in the piece (I know that they are aware of it themselves - probably just taking it for granted that others know too!). Working with social scientists and the humanities requires some flexibility and willingness to manage conflict - but it can hugely enhance physical science. Work with a range of social scientists, and make sure that their skills complement yours – you don't have to be a great communicator if you work with people who specialise in that, and you don't have to have a perfect understanding of development – work with those who do.

---

## Referee Comment (RC2) · Anonymous Referee #2 · 19 Oct 2020

Three words can summarise the submitted manuscript: complete, comprehensive and clear. The authors proposed a very well written perspective on the theme of natural hazards and its new (and old) frontiers. I have a couple of recommendations that the authors can consider to include in their perspective: -Further explore the concept of 'unintended consequences' of NH mitigation or response actions, being discussed widely within the scientific community since a couple of years (see for example Di Baldassarre et al. 2018 https://doi.org/10.1002/2017EF000764); -Further explore the concept of 'prevention', that is always invisibile since as Kofi Annan said in 1999 the benefits of preventive measures are not tangible because they are the disasters that never happened;

---

## Author Comment (AC1) · 23 Oct 2020

**RESPONSE TO REVIEWERS AND SHORT COMMENTS (nhess-2020-163)**

**Invited Perspective: Building sustainable and resilient communities – Recommended actions for natural hazard scientists**

*Joel C. Gill, Faith E. Taylor, Melanie J. Duncan, Solmaz Mohadjer, Mirianna Budimir, Hassan Mdala, and Vera Bukachi*

**[RC1] Anonymous Referee #1**

**Review:** *Gill and co-authors propose a very well-written and interesting perspective on recommended actions for scientists in the context of disaster risk reduction and the sustainable development goals. These recommendations are specifically addressed to researchers involved in the study of natural hazards with the aim that their engagement in DRR and sustainable development is strengthened. What Gill and co-authors propose is timely and the proposed seven recommendations nicely encompass the whole context behind the study of natural hazards and the goal to achieve DRR. The authors have done a very good job of using a diverse literature that covers various natural hazards.*

*I have made few minor comments on the PDF that could improve the manuscript, which I will not repeat here. I have also pointed out few typos. Please also note the supplement to this comment:* [https://nhess.copernicus.org/preprints/nhess-2020-163/nhess-2020-163-RC1-supplement.pdf](https://nhess.copernicus.org/preprints/nhess-2020-163/nhess-2020-163-RC1-supplement.pdf)

**Response:** Thank you to the anonymous referee for this very positive and helpful review. To address the specific comments made:

*[RC1a] the community of natural hazard scientists is identified through a list of diverse disciplines in geoscience. In that list I suggest (physical) geographers and geomorphologist be added.*

[**Response to RC1a**] – We agree and will broaden the list in Section 1 to encompass these extra disciplines.

*[RC1b] among the new approaches to data collection; mention of citizen sciences could be done. Citizen-based approaches are also allow to improve access to hazards information (2.5) and also favour access to marginalized groups and better engagement with indigenous communities (2.6). Citizen-based approaches are also a good way to involve and/or interact with stakeholders (2.3) and better understand the cultural context (2.4).*

*Examples of citizen literature:*

- *Hicks A, Barclay J, Chilvers J, et al (2019) Global Mapping of Citizen Science Projects for Disaster Risk Reduction. Frontiers in Earth Science 7:226. doi: 10.3389/feart.2019.00226*
- *Jacobs L, Kabaseke C, Bwambale B, et al (2019) The geo-observer network: A proof of concept on participatory sensing of disasters in a remote setting. Science of The Total Environment 670:245–261. doi: 10.1016/j.scitotenv.2019.03.177*

[**Response to RC1b**] – We agree that citizen science is important to include and thank the reviewer for identifying this omission. To embed this approach into our manuscript, we will (i) introduce citizen science in **Section 2.1**, and then refer back to its benefits in further sections in **Section 2** as suggested by the reviewer.

*[RC1c] In the Global South, long-term partnerships is often implying capacity-building of local researchers (through a.o. PhD fellowships). However, funding opportunities and grant applications for fundamental research are not always designed to involve a capacity-building component in their projects. Maybe something could be said about it.*

[**RC1c**] This is an interesting point, and we certainly agree that long-term partnerships are an excellent platform for 2-way capacity strengthening, with everybody growing in their knowledge and ability to deploy

appropriate skills in different contexts. Building these partnerships requires funding (highlighted in the Suggested Actions/Priorities for Change in **Section 2.2**), and we propose adding an extra sentence here to note the additional benefits to capacity strengthening work. We also agree that *'funding opportunities and grant applications for fundamental research are not always designed to involve a capacity-building component in their projects'*, although note that this does not prevent an approach being adopted by researchers that allows for learning and development by all.

**[RC2] Anonymous Referee #2**

**Review:** *Three words can summarise the submitted manuscript: complete, comprehensive and clear. The authors proposed a very well written perspective on the theme of natural hazards and its new (and old) frontiers. I have a couple of recommendations that the authors can consider to include in their perspective: -Further explore the concept of 'unintended consequences' of NH mitigation or response actions, being discussed widely within the scientific community since a couple of years (see for example Di Baldassarre et al. 2018 https://doi.org/10.1002/2017EF000764); -Further explore the concept of 'prevention', that is always invisible since as Kofi Annan said in 1999 the benefits of preventive measures are not tangible because they are the disasters that never happened.*

**Response:** Thank you to the anonymous referee for this very positive and helpful review. To address the comments made with regards *unintended consequences,* we propose:

(i) Including in **Section 2.3** on Theory of Change, how this approach can be used to help identify potential unintended consequences of natural hazards mitigation/response, to inform decision making.

(ii) Adding a sentence on understanding and reducing unintended negative consequences of natural hazards mitigation/response to **Section 4**, where we cover benefits of addressing the priorities for change set out in **Section 2**. Many of the issues we raise help to contribute to this aim (e.g., through understanding cultural context, ensuring a greater range of voices are involved in decision making).

To address the comments made with regards *'prevention',* we propose:

(iii) We suggest that a detailed analysis of the extent to which the benefits of preventive measures are or are not tangible is beyond the scope of this paper, but we do agree that it is helpful to make reference to this. We propose doing this in the context of **Section 2.2**, where we discuss the benefits of working with those at risk (such partnerships could help to make preventive measures more tangible to those at risk), and in **Section 2.7** where we note the links between DRR and sustainable development (highlighting how the integration of DRR into sustainable development policy and practice could help decision makers recognise the benefits of 'intangible' prevention measures).

**[SC1] Comment by Amy Donovan**

**Comment**: *This is a great paper and makes some excellent points around the need for natural scientists to be trained in a much wider range of methods and approaches. One thing that I think is there but needs greater emphasis is the need for interdisciplinary working - you mention it multiple times, but there does not seem to be a lot of detail on the complexity of it or the expertise that is available from other disciplines to supplement that of natural scientists. It isn't necessary for natural scientists to become social scientists - the social sciences have a wide range of experience and expertise that is not very well respected a lot of the time in the natural/physical science community. This is NOT just about "behavioural science" (which is a narrow field with very dubious epistemic foundations some of the time) - it also requires working with environmental social scientists (who use mixed methods), anthropologists and human geographers. A key requirement of such working is flexibility - social science typically involves different assumptions about what*

*makes robust knowledge than the physical sciences do, and so awareness of and respect for that is of critical importance. In summary - I'd just encourage the authors to think about this in a bit more detail in the piece (I know that they are aware of it themselves - probably just taking it for granted that others know too!). Working with social scientists and the humanities requires some flexibility and willingness to manage conflict - but it can hugely enhance physical science. Work with a range of social scientists, and make sure that their skills complement yours – you don't have to be a great communicator if you work with people who specialise in that, and you don't have to have a perfect understanding of development – work with those who do.*

**Response:** Thank you to Dr Donovan for the positive comments on this invited perspective, and taking the time to share your expertise so as to strengthen our work. We agree with the comments made about strengthening the discussion of interdisciplinarity in this paper, discussing the challenges and range of expertise natural hazard scientists could engage with. We propose adding (i) a paragraph to **Section 2.2** about the need for partnerships across disciplines (particularly those working in a range of social science disciplines) and challenges of doing so, (ii) strengthening **Section 2.4** on cultural understanding to acknowledge the range of social science expertise that would support natural hazard scientists in this work, and (iii) ensuring this is captured in the discussion in **Section 3**.